# Identifying and Prioritising Behaviours to Slow Antimicrobial Resistance

**DOI:** 10.3390/antibiotics12060949

**Published:** 2023-05-23

**Authors:** Fraser Tull, Rebecca S. Bamert, Liam Smith, Denise Goodwin, Karen Lambert

**Affiliations:** 1BehaviourWorks Australia, Monash Sustainable Development Institute, Monash University, Melbourne 3170, Australia; fraser.tull@monash.edu (F.T.); liam.smith@monash.edu (L.S.); denise.goodwin@monash.edu (D.G.); 2Centre to Impact AMR, Monash University, Melbourne 3170, Australia; rebecca.bamert@monash.edu; 3Faculty of Education, Monash University, Melbourne 3170, Australia

**Keywords:** antimicrobial resistance, AMR, behaviour change, impact-likelihood

## Abstract

As a nation with relatively low levels of AMR, due to both community and agricultural stewardship, as well as geographical isolation, Australia is somewhat unique. As this advantage is being eroded, this project aimed to investigate the spectrum of human behaviours that could be modified in order to slow the spread of AMR, building upon the argument that doable actions are the best-targeted and least complex to change. We conducted a workshop with a panel of diverse interdisciplinary AMR experts (from sociology, microbiology, agriculture, veterinary medicine, health and government) and identified twelve behaviours that, if undertaken by the public, would slow the spread of AMR. These were then assessed by a representative sample of the public (285 Australians) for current participation, likelihood of future participation (likelihood) and perceived benefits that could occur if undertaken (perceived impact). An impact-likelihood matrix was used to identify four priority behaviours: do not pressure your doctor for antibiotics; contact council to find out where you can safely dispose of cleaning products with antimicrobial marketing; lobby supermarkets to only sell antibiotic free meat products; and return unused antibiotics to a pharmacy. Among a multitude of behavioural options, this study also highlights the importance of tailoring doable actions to local conditions, increasing community education, and emphasizing the lack of a one-size fits all approach to tackling this global threat.

## 1. Introduction

The unfolding threat of a future in which standard medical procedures and everyday infections once again become life-threatening events is edging ever-closer, due to antimicrobial resistance (AMR). Antimicrobial resistance is the ability of bacteria, viruses and fungi to subvert the drugs used to kill them, and results in infections that are unable to be cleared. As such, the World Health Organisation (WHO) has declared AMR one of its top global health priorities [1]. Despite this declaration and a growing scientific awareness of the problem, resistance is still increasing globally (and locally). In fact, the COVID pandemic reversed years of incremental progress in decreasing antimicrobial use in the US, was associated with a 15% increase in hospital-related deaths and infections in 2020 [2] and is accelerating our progress toward a post-antibiotic future [3].

Globally investment is being made in new drugs, however, without other interventions these will simply result in new resistances within ever decreasing timeframes. Given a CDC study showing that 30% of antibiotic prescriptions in the US are inappropriate [4], medical systems, practitioners and practices are increasingly well-studied and targeted for intervention. Beyond stewardship, changing our built and natural environment, reversing evolution, and investing in basic research on microbes and biomes are also part of the solutions being proposed. Often overlooked is the role of individuals and their behaviour. Indeed, behaviour is at the centre of the AMR problem [5], and without behaviour change it is unlikely that medical solutions alone will be sufficient to prevent increased AMR and the ensuing health threats. Despite this, much of the public narrative surrounding AMR (discovery-based scientific breakthrough reporting) only reinforces to the community that this is a science problem and not a behavioural problem [6].

In Australia in 2019, 1031 deaths were directly attributable to antibiotic-resistant bacterial infections, of a total of ~22000 AMR infections that resulted in hospital stays [7]. While Australia currently stands in good stead globally with respect to AMR, with only 6.5 deaths per 100,000 directly attributable to AMR, compared with 16.5 per 100,000 globally [8], it is important that countries such as Australia continue to pursue interventions to ameliorate, and even reverse, the rise in AMR. As suggested in this paper, one area where more attention is needed is an increased role for the public in the prevention of AMR. This study aims to fill this gap by gauging public response and amenability to actions to this abstract, yet growing, threat. 

In some regions, behaviour-change approaches with respect to AMR have been explored, often times focusing on prescriber behaviour more than consumer behaviour [9]. Ultimately, though, changing both clinician behaviour and patient behaviour is required [10]. With a focus on antibiotic medicine use, comparative approaches to the controls used in Sweden, UK and Australia found that Sweden has significantly more regulatory control, compared to Australia and the UK, where both practitioner and public education have been the focus [11]. The authors concluded that Sweden’s approach has largely been more effective, as evidenced by reported knowledge, attitudes and behaviours in Australia and the UK, which are each less consistent with appropriate antibiotic use than in Sweden [11].

However, one model of behaviour change (RESET) posits that regulation is just one of the levers, and that intrinsically motivated change (based upon education and social pressure) can induce more lasting change that would require less monitoring in the long term. This model was successfully applied to discourage the use of antibiotics in the Netherlands dairy industry, demonstrating its applicability to the problem of AMR [12]. 

In general, while using behavioural models to understand behaviour and design interventions is laudable, an important first step is to identify which behaviours to target in the first place. Any successful policy or behaviour-change campaign begins with knowing which interventions to target [13]. Ostensibly, there are several behaviours that, if undertaken en masse, would greatly assist in a reduction of AMR, including several that go beyond altering the prescription and use of medicinal antibiotics, but understanding these in a local context is key to genuine community engagement and eventual success [14]. For example, engaged communities could avoid buying items that contain antibacterial agents, lobby governments for greater control of AMR use or ensure that unused antibiotic products are disposed of correctly. 

Given that many public policy issues such as AMR involve community behaviour change, and that behavioural science is more effective when focused on particular behaviours [15], most behaviour-focused research processes advocate and include one or more steps to identify and prioritise behaviour. For example, the UK’s Behavioural Insights Team suggest a phase of identifying and prioritising behaviours based on impact and feasibility [16]; the BASIC model [17] includes a behavioural reduction tool to identify behaviours in concert with stakeholders and poses several questions to filter and prioritise behaviour. In line with these tools, Kneebone et al. [18] developed an impact-likelihood model which uses assessments of impact (the impact of the behaviour on the issue), likelihood of uptake (based on perceived ease) and current level of participation (the number of people already performing the behaviour). These tools align with the recommendation that AMR actions which are beneficial and doable are the best targeted [19].

Given the need for behaviour change to address AMR, a reticence to use regulatory tools in Australia [11] and the need for community behaviour change, this project aimed to investigate the spectrum of human behaviours that could be modified in order to slow the spread of AMR. This paper first explores audiences that would be a suitable target for a behaviour change intervention and a suite of behaviours for each audience. Following this, we seek to identify specific behaviours that could be suitable targets for intervention based on an application of the impact-likelihood matrix.

The following research questions were examined:RQ1. What audience/actor is best to target for behaviour change interventions, based on the potential of their behaviour to impact AMR?RQ2. Which public behaviours are likely to have the greatest impact on slowing AMR?RQ3. What are the current adoption rates for each of these behaviours (and therefore, which behaviours offer the greatest opportunity for changing behaviour)?RQ4. Which behaviours are the public most likely to engage in?RQ5. Where do their behaviours sit within a behavioural selection tool that assesses impact, likelihood and current participation (i.e., the impact-likelihood matrix)?

## 2. Results

### 2.1. Behaviour Identification

In order to answer RQs 1 and 2, an expert panel (*n* = 12) identified relevant behaviours to target amongst the general public within Australia, in order to slow AMR. This was followed by an immediate ranking of each of the behaviours in terms of their potential impact on slowing AMR, resulting in 12 behaviours that were identified as potentially impactful. These then became the focus of this research (Appendix A). The 12 behaviours fell into four groups; antibiotics use, lobbying for change, antimicrobial marketing, and regular hand-washing. 

Three of the community behaviours related to the human use of antibiotics. These included: not “pressuring” your doctor for antibiotics; only taking antibiotics when prescribed by a doctor; and returning unused antibiotics to a pharmacy. 

Five of the community behaviours related to antimicrobial marketing on cleaning products (e.g., cleaning products with marketing on their package promoting antimicrobial agents—“plus antimicrobial”, “plus antibacterial” or “kills 99.9% of germs”). These behaviours were: choosing personal cleaning products (e.g., hand soaps and body wash) that do not promote antimicrobial agents; choosing household cleaning products (e.g., multi-purpose cleaner and dishwashing liquid) that do not promote antimicrobial agents; encouraging people close to you (e.g., family and friends) to choose personal and household cleaning products that do not promote antimicrobial agents; asking workplace procurement to choose personal and household cleaning products that do not promote antimicrobial agents; and contacting the local council to find out where you can safely dispose of personal or household cleaning products that do promote antimicrobial agents.

Three of the community behaviours related to lobbying for change. These included: lobbying the regulator, or product manufacturer, against antimicrobial marketing; lobbying the regulator, or product manufacturer, to remove antimicrobial agents from their cleaning products; and lobbying supermarkets to only sell “antibiotic free” meat products. 

One community behaviour related to stopping the spread of bacteria: regular handwashing. 

### 2.2. Behaviour Impact

A follow up survey with a broader expert panel (*n* = 21) resulted in an impact assignment for each of the 12 behaviours. The experts generally felt that the most impactful behaviour was, “do not pressure your doctor for antibiotics” and the least impactful behaviour was, “choose personal cleaning products without antimicrobial marketing”. Other behaviours that were seen to have greater impact included “contact council to find out where you can safely dispose of products with antimicrobial marketing” and “lobby to remove antimicrobial agents from cleaning products”.

### 2.3. Behaviour Adoption

The third phase of the study involved taking these behaviours to the general public (*n* = 285) for assessment of current and potential behaviour adoption. Adoption rates varied considerably across behaviours, ranging from 2.46% “lobby to remove antimicrobial agents from cleaning products” to 95.44% “washing your hands regularly”. Six behaviours had less than 10% of the sample currently engaging in the behaviour, and a further three behaviours had less than 34% of participants engaging in the behaviour, indicating considerable opportunity for behaviour change across most of the behaviours. Behaviours that offered less opportunity for behaviour change included “washing your hands regularly”, “only take antibiotics when prescribed”, and “do not pressure your doctor for antibiotics”.

### 2.4. Behaviour Likelihood

Participants indicated that, moving forward, they were likely to engage in eight of the twelve behaviours, which is to say, they tended to agree with the statements. Those behaviours that were most likely to be adopted were “washing your hands regularly” and “only take antibiotics when prescribed”. Those behaviours that were not likely to be adopted included “lobby to remove antimicrobials from cleaning products”, “lobby against antimicrobial marketing on cleaning products”, “at your workplace, ask procurement to choose products without antimicrobial marketing”, and “encourage the people close to you to choose products without antimicrobial marketing”.

### 2.5. Impact-Likelihood Matrix

As shown in Figure 1, below, when mapping the behaviours onto the matrix it becomes clear that some behaviours are better targets for behaviour-change intervention than others. Percentage value and circle size indicate the percentage of participants who are already engaging in the behaviour, among those who have had the opportunity to perform the behaviour. (Larger circles indicate higher levels of behaviour-adoption, and therefore less opportunity to intervene and change behaviour). Green circles denote direct antibiotic use behaviours, yellow circles lobbying behaviours, blue circles are marketing/purchasing behaviours and grey is hand hygiene.

Two behaviours fell into the top right quadrant (higher impact, higher likelihood). These included “only take antibiotics when prescribed” and “do not pressure your doctor for antibiotics”. The first of these behaviours is already being undertaken by the vast majority of people and so offers little room for improvement. However, there does appear to be sufficient room to further encourage people to avoid asking their doctor for antibiotics.

Five behaviours fell into the top left quadrant (higher impact, lower likelihood). These included all of the lobbying behaviours and “at your workplace, ask procurement to choose products without antimicrobial marketing” and “contact council to find out where you can safely dispose of products with antimicrobial marketing”. All of these behaviours had extremely low adoption rates and offered considerable room for behaviour change. However, only two of these behaviours had likelihood scores that afforded any real potential for future adoption. These included, “contact council to find out where you can safely dispose of products with antimicrobial marketing” and “lobby supermarkets to only sell antibiotic-free meat products”.

Two behaviours fell into the bottom left quadrant (lower impact, lower likelihood). These included “encourage the people close to you to choose products without antimicrobial marketing” and “choose household cleaning products without antimicrobial marketing”. Both of these behaviours had either low or very low current adoption rates suggesting considerable room for behaviour change. However, with their lower impact and lower likelihood scores they do not appear to be suitable targets for intervention.

Three behaviours fell into the bottom right quadrant (lower impact, higher likelihood). The first was “washing your hands regularly”, however this behaviour had extremely high current adoption rates offering little room to change behaviour. The second was, “choose personal cleaning products without antimicrobial marketing”, but this was the lowest-ranked behaviour in terms of impact across all of the behaviours. The third behaviour was, “returning unused antibiotics to a pharmacy” which tends to have sufficient room for behaviour change and reasonable impact and likelihood of future adoption.

Based on their position within the matrix, along with their current adoption rates, the following behaviours appear to offer the best opportunity to target through intervention:A.Do not pressure your doctor for antibiotics;B.Contact council to find out where you can safely dispose of products with antimicrobial marketing;C.Return unused antibiotics to a pharmacy;D.Lobby supermarkets to only sell antibiotic-free meat products.

### 2.6. Consumer Engagement with AMR

Encouragingly, the final question (designed to explore the extent to which a general audience is interested in learning more about AMR) showed that 39.79% of participants were interested in finding out more about AMR. Within a general-consumer panel sample this represents a high level of engagement. Comments within the general feedback question also indicated that people were largely unaware of the topic, but thought it was important.

## 3. Discussion

The above-identified behaviours echo the breadth of the initial responses from the assembled twelve AMR experts, and the difficulty of tackling AMR from only one angle.

Antibiotic stewardship featured heavily in the upper right quadrant, and with the highest percentages of participants already undertaking those behaviours; clearly messaging about antibiotic overuse is starting to break through. Recent studies have confirmed that ‘pressuring doctors for antibiotics’ (A) is not as widespread a behaviour as anecdotally assumed [20] and may be contributing less to the continued spread of AMR in Australia, a country without the potential demands associated with direct-to-consumer marketing of pharmaceuticals [21]. This must be tempered by the acknowledgement that self-reporting (especially following a presentation on the harms of the activity) can be subject to bias [16], or possibly masked by a lack of understanding of what medications may actually be antimicrobials [20,22]. Less subjectively, community use of antimicrobials remains very high in Australia, with per-capita usage ranking seventh-highest when ranked against European nations [23], albeit slowly declining. Hence, there may be some room, even amongst this cohort, to stop pressuring doctors for antibiotics. This would have a high impact at a personal level, as one facet of targetable interactions at the physician–patient interface [24,25] and at the local, Australian level. More so (and beyond the scope of this study) would be the opportunity to spread the message across parents and pet owners not to seek antibiotics unnecessarily for children and animals in their care.

In the broader research-centred context of AMR, the One Health framework is viewed as a cornerstone of any successful approach to combating the global rise in AMR [26]. One Health is the concept that optimal health outcomes rely upon acknowledgment of the interconnection between animals, plants, people and the environment [27]. Seen through this lens, the behaviours linked to disposal of unused medication and cleaning products (B) are worth exploring. Correctly disposing of materials that contain antimicrobials prevents them from entering the environment, soil and water as pollutants, and possibly becoming triggers of resistance in the bacteria that reside there [28]. Prevention of AMR in these environmental reservoirs prevents transmission of these AMR genes or organisms to humans through the food chain, water or environmental exposure. Adoption of two of the behaviours identified (B and C) would divert antimicrobial compounds from the environment; however, given that usage of these pathways relies upon convenience and familiarity [29], more work needs to be performed to make this behaviour accessible and simple for consumers. Even amidst a population that was very supportive of household waste sorting schemes, there were divisions around usage of local (or distant) drop-off points for hazardous waste [30], and hesitations relating to the time cost of such processes. In an Australian context, information from local councils on hazardous waste disposal (including unwanted antimicrobial cleaning products) is generally readily available, although schemes vary in their ease of access and use. For example, in the City of Greater Melbourne, councils tend to employ an external provider with moving pickup points, low frequency of collection and online pre-registration for users.

In terms of correct disposal of antibiotic medications (C), the National Return and Disposal of Unwanted Medicines (RUM) Project [31] has been operational in Australia since 1996 and has seen year-to-year increases in total medicines collected and correctly disposed of (https://returnmed.com.au/collections/, accessed on 12 March 2023). An audit of returned medicines regularly shows antibiotics within the top 20 most commonly dispensed and returned prescription medicines in Australia [32], despite many consumers stating that antibiotics are a class of drug of which they would keep an excess, ‘just in case’ [33]. Once informed of this service (often by pharmacists), consumers are willing to return unwanted medicines [34]. Coupled with education about the risk of AMR resulting from improperly used and/or disposed of antibiotics, a simple behaviour change could be associated with a real impact on the amount of antibiotics entering landfill or wastewater.

The One Health concept also extends to farming, but specifically, in this instance, the use of antibiotics in meat production is targeted as another behaviour change (D). Concern about both the use of antibiotics in food and the ready transmission of antibiotic residues or AMR pathogens to the consumer is reflected in the call to lobby supermarkets to only sell antibiotic-free meat. In the Australian context, the use of antibiotics in meat production is relatively limited, and coupled with the National Residue Survey testing of foods sold for human consumption (National Residue Survey Administration Act 1992), not a problem likely to be contributing significantly to Australian AMR levels. However, for other countries where stewardship of antimicrobials in the food production industry is less stringent, this provides a good target for change, as people are generally highly motivated to avoid ‘contaminants’ in foods. Interestingly, these disconnects between the actual and perceived presence of antibiotics in Australian meat and in Australian agricultural practices again highlight a gap in knowledge and education.

Encouragingly, this gap is one that general consumers have an interest in filling. Although not the primary focus of this research, we also explored to what degree the general public was interested in learning more about the topic. Almost 40% of general-consumer participants wished to know more about AMR, representing a high level of engagement and a potential for education and behaviour changes to be adopted. This receptivity reflects the fact that consumers commented that, despite previously knowing little about the topic, they felt it was important. Encouragingly, research has shown that increased understanding of AMR does correlate with decreased inappropriate use of antibiotics, underscoring the value of targeted education campaigns [25].

By targeting the correct behaviours for a local population and communicating clearly, inroads into reversing the rising tide of antimicrobial resistance can be made [35,36]. The current study suggests that people are engaged with the topic, interested, and can understand that it can potentially affect their health and that of their families. These attitudes make it more likely that tailored behaviour change will be adopted.

## 4. Materials and Methods

The process of behaviour identification, prioritisation, impact, adoption and likelihood was broken into three main phases: an initial workshop to identify and prioritise public behaviour; a follow up survey with workshop participants to accomplish the same (but with a focus on prioritisation and more time to consider); and a survey of the general public to capture the likelihood of participation.

### 4.1. Behaviour Identification

In order to answer RQs 1 and 2 (i.e., identify a suitable target actor and their behaviours), a 4 h online behaviour identification workshop was conducted with 12 AMR experts across Australia and New Zealand, in April 2022. The experts were drawn from the authors’ contacts in the fields of sociology, microbiology, public health and government, following consultation with Monash Centre to Impact AMR’s Director, management, executive and transdisciplinary group leader cohorts. The scope of the workshop was focused on consumers (e.g., the general public) rather than providers (e.g., doctors) while recognising that some groups may play both roles (e.g., farmers). The experts participated in a system-mapping exercise to identify the various groups of people who play a role in slowing AMR, and subsequently undertook a prioritization process to narrow the list down to two or three target groups and their relevant behaviours. The experts then completed a survey comparing each of the behaviours in terms of their potential impact on slowing AMR. Overall, most of the behaviours that were rated as potentially having the greatest impact pertained to the general public, so this group, along with the 12 behaviours that were identified as potentially impactful, became the focus of this research (Appendix A). The 12 behaviours fell into four groups; antibiotics use, lobbying for change, antimicrobial marketing and regular hand washing. 

### 4.2. Behaviour Impact 

After the behaviour identification workshop, the next phase of this research involved conducting an expert survey with a broader group of AMR experts (*n* = 21) to answer RQ2 (i.e., identify the relative potential impact of the 12 behaviours). Experts known by the Centre to Impact AMR were emailed an invitation to complete the online survey in June 2022. Participants were from academia (*n* = 17), government (*n* = 2) and industry (*n* = 2), with between 1 and 26 years (M = 7.5 years) experience working in the area of AMR. When completing the survey, participants were presented with a task to rank the twelve behaviours from “1”, most impactful, to “12”, least impactful, in terms of their potential impact on slowing AMR if a person were to adopt the behaviour whenever possible. The order in which the behaviours were presented in the list was randomized across participants. In order to calculate an impact score for each behaviour, each participant’s score was reverse-coded (e.g., a value of 1 received a score of 12), and then the average was taken across the sample, so higher scores indicate greater relative impact.

### 4.3. Behaviour Adoption and Likelihood 

Data for RQ3 (i.e., current behaviour adoption) and RQ4 (i.e., likelihood of future adoption) came from 285 adults who completed an online survey in June 2022. A survey is a common research method used to collect data from a sample of individuals or groups; they are widely used in various fields, including social sciences, market research and public health. Surveys involve asking a series of structured questions, in our case in an online format, to elicit specific information related to our research topic. Respondents were randomly selected by a research company from their panel of members with quotas set for age, gender and geographic location, so that the sample broadly reflected the Australian adult population. From the initial sample, 63 participants were excluded due to suspected speeding as indicated by taking less than two seconds to respond to a question on a new page, across at least one quarter of key pages in the survey. The final sample comprised 52.63% female members, 75.44% of whom were living in a metropolitan area, with an average age of 51.78 years (SD = 18.48 years). 

Due to the anticipated lack of public knowledge on the topic, participants were first presented with a brief description of AMR. Furthermore, due to the large number of behaviours pertaining to antimicrobial marketing on cleaning products, participants were also presented with a description and example of this marketing. Participants were then presented with questions capturing demographic information, followed by twelve pages (one page per behaviour), each containing a description of the behaviour and questions capturing behaviour adoption and likelihood. The behaviours were presented in random order across participants.

To capture behaviour adoption, for low frequency behaviours, participants were asked if they had ever performed the behaviour (e.g., Have you ever signed a petition to … Yes/No), while for higher frequency behaviours participants were asked if they usually perform the behaviour (e.g., Do you usually choose personal cleaning brands that … Yes/No). Some behaviours also included a response option of not applicable (e.g., I do not purchase cleaning products). Participants who selected this option were not included in the analysis for that behaviour. Some behaviours also included a response option of “don’t know”, which was treated as “No” (does not adopt the behaviour). The adoption rate for each behaviour was then calculated by dividing the number of participants who reported doing the behaviour by the total number of participants who had the opportunity to adopt the behaviour.

To capture likelihood, participants were presented with two statements, and for each statement, were asked how much they agreed or disagreed with the statement, from 1, “Strong disagree”, to 5, “Strongly agree”. The first statement captured perceived ease (e.g., It would be easy for me to …), while the second statement captured perceived likelihood (e.g., I am likely to …). An overall measure of likelihood was then calculated for each participant by adding the two scores together, resulting in a score from a low of 2 (very low likelihood) to a high of 10 (very high likelihood). Some behaviours also included a response option of not applicable (e.g., I do not purchase cleaning products). Participants who selected this option were not included in the analysis for that behaviour. Due to a number of behaviours exhibiting a strong skew across the sample, the median was used to calculate a likelihood score for each behaviour among participants who had the opportunity to adopt the behaviour.

### 4.4. Impact-Likelihood Matrix

The findings from the expert survey and the general public survey were then used to answer RQ5 (i.e., mapping each behaviour onto the matrix). We used an impact-likelihood model for this research because we were seeking to prioritize the behaviours that have the highest potential for positive impacts. This type of model helps researchers and decision makers to prioritize their focus and allocate resources, informing the development of tailored interventions or strategies that specifically target high-impact behaviours with a high likelihood of occurrence, as well as facilitate communication and stakeholder engagement [15].

### 4.5. Public Engagement with AMR Content

Although not the primary focus of this research, we also explored to what degree the general public would be interested in learning more about the topic. If people tended to show an interest, that suggests they may engage with interventions or communications about AMR. 

To help answer this, we included an additional question at the end of the general public survey, after a “general feedback” question. The feedback question served to give the impression that there were no more survey questions. This was important, because the final question then asked whether or not they would like to know more about AMR (selecting “Yes” took them to a further information page, selecting “No” closed the survey). Giving the impression that the survey was already over enabled a more direct measurement of interest in the topic (i.e., they were investing their own time to find out more).

## Figures and Tables

**Figure 1 antibiotics-12-00949-f001:**
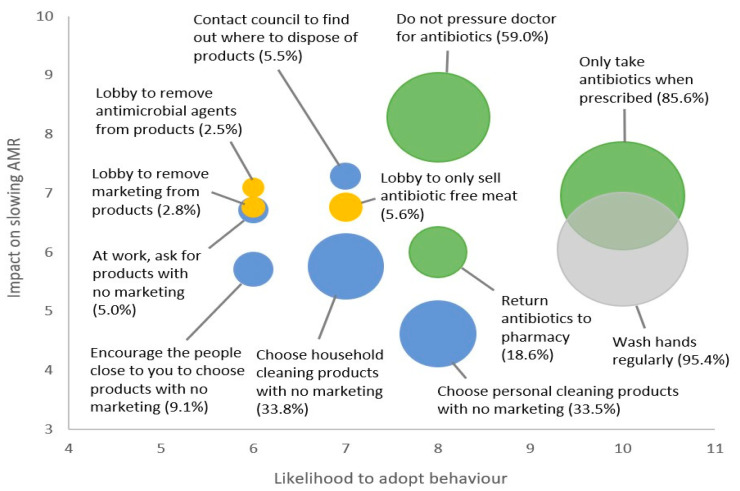
Impact-likelihood matrix of community behaviours to slow AMR.

## Data Availability

The data presented in this study are available in Appendix A
.

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
