# Peer review of "Identifying and Prioritising Behaviours to Slow Antimicrobial Resistance"

_antibiotics, 2023, doi:10.3390/antibiotics12060949_

Round 1

Reviewer 1 Report

  • The findings of the study are interesting and informative. The researchers identified four priority behaviours that could slow the spread of AMR.
  • However, there are a few weaknesses in the study that should be noted.
  • The introduction and discussion sections of a paper should provide a comprehensive overview of the literature on the topic. To do this, it is important to cite a variety of sources, including recent peer-reviewed articles, books, and government reports. By adding more references, you can demonstrate that you have a strong understanding of the current state of knowledge on the topic and that your paper is well-grounded in the literature.
  • Explain method: If your paper is a mixed-methods study, it is important to explain the methods used in both the quantitative and qualitative components of the study. You should also provide a detailed description of the COREQ checklist, which is a set of criteria for evaluating the quality of mixed-methods studies. By providing this information, you can help readers to understand how your study was conducted and to assess the quality of the findings.
  • The study did not assess the effectiveness of the four priority behaviours. It is possible that these behaviours may not be effective in slowing the spread of AMR.
  • Despite these weaknesses, I believe that this study is an important contribution to the field of antimicrobial resistance. The findings of the study can be used to develop strategies to slow the spread of AMR and protect public health.
usechatgpt init success usechatgpt init success

Good

usechatgpt init success usechatgpt init success

Reviewer 2 Report

The study is interesting. I suggest that the final conclusions be summarised in a separate paragraph.

Author Response

REVIEWER: The study is interesting. I suggest that the final conclusions be summarised in a separate paragraph.

RESPONSE: Edited as requested. Thank you for your support.  

Reviewer 3 Report

This manuscript highlights a critical research topic. I enjoyed seeing the use of an impact-likelihood matrix for AMR interventions/high-priority areas. This study is straightforward and valuable for our field. I applaud the authors' hard work engaging with experts and identifying these key messaging areas that will have a broader context for behavior change interventions and educational messaging beyond Europe and Australia. 

I have a few minor comments and suggestions for the authors to consider before accepting this manuscript for publication. 

1) Please, address the relevant literature on this topic in other countries outside Europe and Australia (e.g., the US). Which scholars are investigating antibiotic misuse behaviors? What priority areas have been identified in these places (if applicable)? Also, please justify conducting these expert-panel workshops and surveys. Also, justify using the impact-likelihood model in this context.

2) I missed information on how the expert panel was selected for the workshop (n=12). How were these AMR experts chosen, and what sectors were they coming from? On a similar note, it will be helpful for readers to be directed to a table that shows the sociodemographic characteristics of surveyed participants (in addition to being outlined in the text). 

3) I missed, in the supplement, the questionnaire(s)/survey(s) used during the workshop, post-workshop, and provided to the general public. This information will benefit the readership to review and reference your work in future studies.

Overall, this manuscript is well done but will need a review for grammar and copy editing. A few grammatical errors were detected, and a few longer sentences. However, these can be addressed and fixed.
